# Regularization Learning Networks: Deep Learning for Tabular Datasets

**Ira Shavitt**
Weizmann Institute of Science
`irashavitt@gmail.com`

**Eran Segal**
Weizmann Institute of Science
`eran.segal@weizmann.ac.il`

## Abstract

Despite their impressive performance, *Deep Neural Networks* (DNNs) typically underperform *Gradient Boosting Trees* (GBTs) on many tabular-dataset learning tasks. We propose that applying a different regularization coefficient to each weight might boost the performance of DNNs by allowing them to make more use of the more relevant inputs. However, this will lead to an intractable number of hyperparameters. Here, we introduce *Regularization Learning Networks* (RLNs), which overcome this challenge by introducing an efficient hyperparameter tuning scheme which minimizes a new *Counterfactual Loss*. Our results show that RLNs significantly improve DNNs on tabular datasets, and achieve comparable results to GBTs, with the best performance achieved with an ensemble that combines GBTs and RLNs. RLNs produce extremely sparse networks, eliminating up to $99.8\%$ of the network edges and $82\%$ of the input features, thus providing more interpretable models and reveal the importance that the network assigns to different inputs. RLNs could efficiently learn a single network in datasets that comprise both tabular and unstructured data, such as in the setting of medical imaging accompanied by electronic health records. An open source implementation of RLN can be found at `https://github.com/irashavitt/regularization_learning_networks`.

## 1 Introduction

Despite their impressive achievements on various prediction tasks on datasets with distributed representation [14, 4, 5] such as images [19], speech [9], and text [18], there are many tasks in which *Deep Neural Networks* (DNNs) underperform compared to other models such as *Gradient Boosting Trees* (GBTs). This is evident in various *Kaggle* [1, 2], or *KDD Cup* [7, 16, 27] competitions, which are typically won by GBT-based approaches and specifically by its *XGBoost* [8] implementation, either when run alone or within a combination of several different types of models.

The datasets in which neural networks are inferior to GBTs typically have different statistical properties. Consider the task of image recognition as compared to the task of predicting the life expectancy of patients based on electronic health records. One key difference is that in image classification, many pixels need to change in order for the image to depict a different object [25].[1] In contrast, the relative contribution of the input features in the electronic health records example can vary greatly: Changing a single input such as the age of the patient can profoundly impact the life expectancy of the patient, while changes in other input features, such as the time that passed since the last test was taken, may have smaller effects.

We hypothesized that this potentially large variability in the relative importance of different input features may partly explain the lower performance of DNNs on such tabular datasets [11]. One way to overcome this limitation could be to assign a different regularization coefficient to every weight, which might allow the network to accommodate the non-distributed representation and the variability in relative importance found in tabular datasets.

This will require tuning a large number of hyperparameters. The default approach to hyperparameter tuning is using derivative-free optimization of the validation loss, i.e., a loss of a subset of the training set which is not used to fit the model. This approach becomes computationally intractable very quickly.

Here, we present a new hyperparameter tuning technique, in which we optimize the regularization coefficients using a newly introduced loss function, which we term the *Counterfactual Loss*, or $\mathcal{L}_{CF}$. We term the networks that apply this technique *Regularization Learning Networks* (RLNs). In RLNs, the regularization coefficients are optimized together with learning the network weight parameters. We show that RLNs significantly and substantially outperform DNNs with other regularization schemes, and achieve comparable results to GBTs. When used in an ensemble with GBTs, RLNs achieves state of the art results on several prediction tasks on a tabular dataset with varying relative importance for different features.

## 2 Related work

Applying different regularization coefficients to different parts of the network is a common practice. The idea of applying different regularization coefficients to every weight was introduced [23], but it was only applied to images with a toy model to demonstrate the ability to optimize many hyperparameters.

Our work is also related to the rich literature of works on hyperparameter optimization [29]. These works mainly focus on derivative-free optimization [30, 6, 17]. Derivative-based hyperparameter optimization is introduced in [3] for linear models and in [23] for neural networks. In these works, the hyperparameters are optimized using the gradients of the validation loss. Practically, this means that every optimization step of the hyperparameters requires training the whole network and back propagating the loss to the hyperparameters. [21] showed a more efficient derivative based way for hyperparameter optimization, which still required a substantial amount of additional parameters. [22] introduce an optimization technique similar to the one introduced in this paper, however, the optimization technique in [22] requires a validation set, and only optimizes a single regularization coefficient for each layer, and at most 10-20 hyperparameters in any network. In comparison, training RLNs doesn't require a validation set, assigns a different regularization coefficient for every weight, which results in up to millions of hyperparameters, optimized efficiently. Additionally, RLNs optimize the coefficients in the log space and adds a projection after every update to counter the vanishing of the coefficients. Most importantly, the efficient optimization of the hyperparameters was applied to images and not to dataset with non-distributed representation like tabular datasets.

DNNs have been successfully applied to tabular datasets like electronic health records, in [26, 24]. The use of RLN is complementary to these works, and might improve their results and allow the use of deeper networks on smaller datasets.

To the best of our knowledge, our work is the first to illustrate the statistical difference in distributed and non-distributed representations, to hypothesize that addition of hyperparameters could enable neural networks to achieve good results on datasets with non-distributed representations such as tabular datasets, and to efficiently train such networks on a real-world problems to significantly and substantially outperform networks with other regularization schemes.

## 3 Regularization Learning

Generally, when using regularization, we minimize $\tilde{\mathcal{L}}(Z, W, \lambda) = \mathcal{L}(Z, W) + \exp(\lambda) \cdot \sum_{i=1}^{n} \|w_i\|$, where $Z = \{(x_m, y_m)\}_{m=1}^{M}$ are the training samples, $\mathcal{L}$ is the loss function, $W = \{w_i\}_{i=1}^{n}$ are the

weights of the model, $\|\cdot\|$ is some norm, and $\lambda$ is the regularization coefficient,[2] a hyperparameter of the network. Hyperparameters of the network, like $\lambda$, are usually obtained using cross-validation, which is the application of derivative-free optimization on $\mathcal{L}_{CV}(Z_t, Z_v, \lambda)$ with respect to $\lambda$ where $\mathcal{L}_{CV}(Z_t, Z_v, \lambda) = \mathcal{L}\left(Z_v, \arg\min_W \tilde{\mathcal{L}}(Z_t, W, \lambda)\right)$ and $(Z_t, Z_v)$ is some partition of $Z$ into train and validation sets, respectively.

If a different regularization coefficient is assigned to each weight in the network, our learning loss becomes $\mathcal{L}^{\dagger}(Z, W, \Lambda) = \mathcal{L}(Z, W) + \sum_{i=1}^{n} \exp(\lambda_i) \cdot \|w_i\|$, where $\Lambda = \{\lambda_i\}_{i=1}^{n}$ are the regularization coefficients. Using $\mathcal{L}^{\dagger}$ will require $n$ hyperparameters, one for every network parameter, which makes tuning with cross-validation intractable, even for very small networks. We would like to keep using $\mathcal{L}^{\dagger}$ to update the weights, but to find a more efficient way to tune $\Lambda$. One way to do so is through SGD, but it is unclear which loss to minimize: $\mathcal{L}$ doesn't have a derivative with respect to $\Lambda$, while $\mathcal{L}^{\dagger}$ has trivial optimal values, $\arg\min_{\Lambda} \mathcal{L}^{\dagger}(Z, W, \Lambda) = \{-\infty\}_{i=1}^{n}$. $\mathcal{L}_{CV}$ has a non-trivial dependency on $\Lambda$, but it is very hard to evaluate $\frac{\partial \mathcal{L}_{CV}}{\partial \Lambda}$.

We introduce a new loss function, called the *Counterfactual Loss* $\mathcal{L}_{CF}$, which has a non-trivial dependency on $\Lambda$ and can be evaluated efficiently. For every time-step $t$ during the training, let $W_t$ and $\Lambda_t$ be the weights and regularization coefficients of the network, respectively, and let $w_{t,i} \in W_t$ and $\lambda_{t,i} \in \Lambda_t$ be the weight and the regularization coefficient of the same edge $i$ in the network. When optimizing using SGD, the value of this weight in the next time-step will be $w_{t+1,i} = w_{t,i} - \eta \cdot \frac{\partial \mathcal{L}^{\dagger}(Z_t, W_t, \Lambda_t)}{\partial w_{t,i}}$, where $\eta$ is the learning rate, and $Z_t$ is the training batch at time $t$.[3] We can split the gradient into two parts:

$$w_{t+1,i} = w_{t,i} - \eta \cdot (g_{t,i} + r_{t,i}) \tag{1}$$

$$g_{t,i} = \frac{\partial \mathcal{L}(Z_t, W_t)}{\partial w_{t,i}} \tag{2}$$

$$r_{t,i} = \frac{\partial}{\partial w_{t,i}} \left( \sum_{j=1}^{n} \exp(\lambda_{t,j}) \cdot \|w_{t,j}\| \right) = \exp(\lambda_{t,i}) \cdot \frac{\partial \|w_{t,i}\|}{\partial w_{t,i}} \tag{3}$$

We call $g_{t,i}$ the gradient of the empirical loss $\mathcal{L}$ and $r_{t,i}$ the gradient of the regularization term. All but one of the addends of $r_{t,i}$ vanished since $\frac{\partial}{\partial w_{t,i}}(\exp(\lambda_{t,j}) \cdot \|w_{t,j}\|) = 0$ for every $j \neq i$. Denote by $W_{t+1} = \{w_{t+1,i}\}_{i=1}^{n}$ the weights in the next time-step, which depend on $Z_t$, $W_t$, $\Lambda_t$, and $\eta$, as shown in Equation 1, and define the Counterfactual Loss to be

$$\mathcal{L}_{CF}(Z_t, Z_{t+1}, W_t, \Lambda_t, \eta) = \mathcal{L}(Z_{t+1}, W_{t+1}) \tag{4}$$

$\mathcal{L}_{CF}$ is the empirical loss $\mathcal{L}$, where the weights have already been updated using SGD over the regularized loss $\mathcal{L}^{\dagger}$. We call this the Counterfactual Loss since we are asking a counterfactual question: *What would have been the loss of the network had we updated the weights with respect to $\mathcal{L}^{\dagger}$?* We will use $\mathcal{L}_{CF}$ to optimize the regularization coefficients using SGD *while learning the weights of the network simultaneously* using $\mathcal{L}^{\dagger}$. We call this technique Regularization Learning, and networks that employ it *Regularization Learning Networks* (RLNs).

**Theorem 1.** *The gradient of the Counterfactual loss with respect to the regularization coefficient is* $\frac{\partial \mathcal{L}_{CF}}{\partial \lambda_{t,i}} = -\eta \cdot g_{t+1,i} \cdot r_{t,i}$

*Proof.* $\mathcal{L}_{CF}$ only depends on $\lambda_{t,i}$ through $w_{t+1,i}$, allowing us to use the chain rule $\frac{\partial \mathcal{L}_{CF}}{\partial \lambda_{t,i}} = \frac{\partial \mathcal{L}_{CF}}{\partial w_{t+1,i}} \cdot \frac{\partial w_{t+1,i}}{\partial \lambda_{t,i}}$. The first multiplier is the gradient $g_{t+1,i}$. Regarding the second multiplier, from Equation 1 we see that only $r_{t,i}$ depends on $\lambda_{t,i}$. Combining with Equation 3 leaves us with:

$$\frac{\partial w_{t+1,i}}{\partial \lambda_{t,i}} = \frac{\partial}{\partial \lambda_{t,i}} \left( w_{t,i} - \eta \cdot (g_{t,i} + r_{t,i}) \right) = -\eta \cdot \frac{\partial r_{t,i}}{\partial \lambda_{t,i}} =$$

$$= -\eta \cdot \frac{\partial}{\partial \lambda_{t,i}} \left( \exp\left(\lambda_{t,i}\right) \cdot \frac{\partial \|w_{t,i}\|}{\partial w_{t,i}} \right) = -\eta \cdot \exp\left(\lambda_{t,i}\right) \cdot \frac{\partial \|w_{t,i}\|}{\partial w_{t,i}} = -\eta \cdot r_{t,i}$$

$\square$

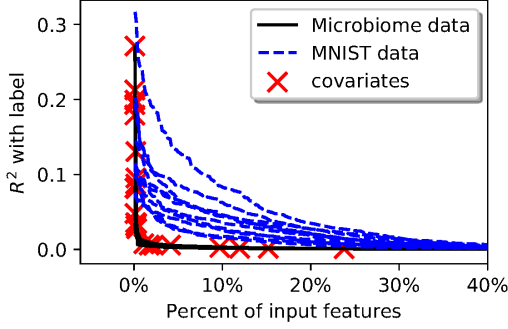

Figure 1: The input features, sorted by their $R^2$ correlation to the label. We display the microbiome dataset, with the covariates marked, in comparison the MNIST dataset[20].

Theorem 1 gives us the update rule $\lambda_{t+1,i} = \lambda_{t,i} - \nu \cdot \frac{\partial \mathcal{L}_{CF}}{\partial \lambda_{t,i}} = \lambda_{t,i} + \nu \cdot \eta \cdot g_{t+1,i} \cdot r_{t,i}$, where $\nu$ is the learning rate of the regularization coefficients.

Intuitively, the gradient of the Counterfactual Loss has an opposite sign to the product of $g_{t+1,i}$ and $r_{t,i}$. Comparing this result with Equation 1, this means that when $g_{t+1,i}$ and $r_{t,i}$ agree in sign, the regularization helps reduce the loss, and we can strengthen it by increasing $\lambda_{t,i}$. When they disagree, this means that the regularization hurts the performance of the network, and we should relax it for this weight.

The size of the Counterfactual gradient is proportional to the product of the sizes of $g_{t+1,i}$ and $r_{t,i}$. When $g_{t+1,i}$ is small, $w_{t+1,i}$ does not affect the loss $\mathcal{L}$ much, and when $r_{t,i}$ is small, $\lambda_{t,i}$ does not affect $w_{t+1,i}$ much. In both cases, $\lambda_{t,i}$ has a small effect on $\mathcal{L}_{CF}$. Only when both $r_{t,i}$ is large (meaning that $\lambda_{t,i}$ affects $w_{t+1}$), and $g_{t+1,i}$ is large (meaning that $w_{t+1}$ affects $\mathcal{L}$), $\lambda_{t,i}$ has a large effect on $\mathcal{L}_{CF}$, and we get a large gradient $\frac{\partial \mathcal{L}_{CF}}{\partial \lambda_{t,i}}$.

At the limit of many training iterations, $\lambda_{t,i}$ tends to continuously decrease. We try to give some insight to this dynamics in the supplementary material. To address this issue, we project the regularization coefficients onto a simplex after updating them:

$$\widetilde{\lambda}_{t+1,i} = \lambda_{t,i} + \nu \cdot \eta \cdot g_{t+1,i} \cdot r_{t,i} \tag{5}$$

$$\lambda_{t+1,i} = \widetilde{\lambda}_{t+1,i} + \left( \theta - \frac{\sum_{j=1}^{n} \widetilde{\lambda}_{t+1,j}}{n} \right) \tag{6}$$

where $\theta$ is the normalization factor of the regularization coefficients, a hyperparameter of the network tuned using cross-validation. This results in a zero-sum game behavior in the regularization, where a relaxation in one edge allows us to strengthen the regularization in other parts of the network. This could lead the network to assign a modular regularization profile, where uninformative connections are heavily regularized and informative connection get a very relaxed regularization, which might boost performance on datasets with non-distributed representation such as tabular datasets. The full algorithm is described in the supplementary material.

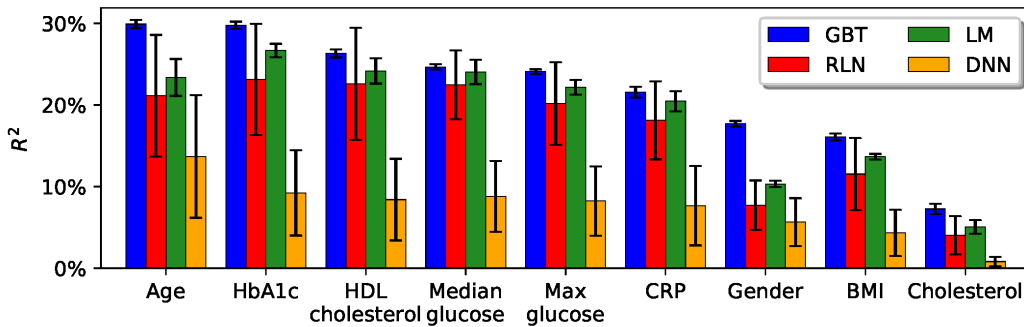

Figure 2: Prediction of traits using microbiome data and covariates, given as the overall explained variance ($R^2$).

## 4 Experiments

We demonstrate the performance of our method on the problem of predicting human traits from gut microbiome data and basic covariates (age, gender, BMI). The human gut microbiome is the collection of microorganisms found in the human gut and is composed of trillions of cells including bacteria, eukaryotes, and viruses. In recent years, there have been major advances in our understanding of the microbiome and its connection to human health. Microbiome composition is determined by DNA sequencing human stool samples that results in short (75-100 basepairs) DNA reads. By mapping these short reads to databases of known bacterial species, we can deduce both the source species and gene from which each short read originated. Thus, upon mapping a collection of different samples, we obtain a matrix of estimated relative species abundances for each person and a matrix of the estimated relative gene abundances for each person. Since these features have varying relative importance (Figure 1), we expected GBTs to outperform DNNs on these tasks.

We sampled 2,574 healthy participants for which we measured, in addition to the gut microbiome, a collection of different traits, including important disease risk factors such as cholesterol levels and BMI. Finding associations between these disease risk factors and the microbiome composition is of

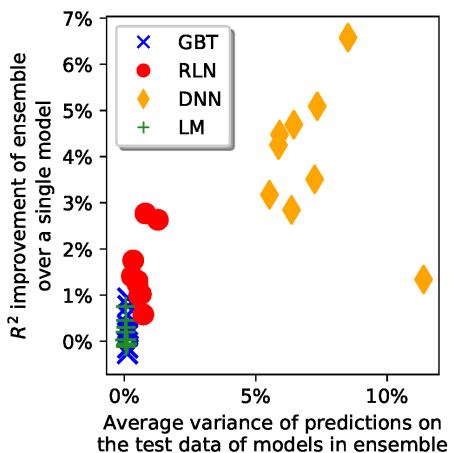

Figure 3: For each model type and trait, we took the 10 best performing models, based on their validation performance, and calculated the average variance of the predicted test samples, and plotted it against the improvement in $R^2$ obtained when training ensembles of these models. Note that models that have a high variance in their prediction benefit more from the use of ensembles. As expected, DNNs gain the most from ensembling.

great scientific interest, and can raise novel hypotheses about the role of the microbiome in disease. We tested 4 types of models: RLN, GBT, DNN, and *Linear Models* (LM). The full list of hyperparameters, the setting of the training of the models and the ensembles, as well as the description of all the input features and the measured traits, can be found in the supplementary material.

## 5 Results

When running each model separately, GBTs achieve the best results on all of the tested traits, but it is only significant on 3 of them (Figure 2). DNNs achieve the worst results, with $15\% \pm 1\%$ less explained variance than GBTs on average. RLNs significantly and substantially improve this by a factor of $\mathbf{2.57 \pm 0.05}$, and achieve only $2\% \pm 2\%$ less explained variance than GBTs on average.

Constructing an ensemble of models is a powerful technique for improving performance, especially for models which have high variance, like neural networks in our task. As seen in Figure 3, the average variance of predictions of the top 10 models of RLN and DNN is $1.3\% \pm 0.6\%$ and $14\% \pm 3\%$ respectively, while the variance of predictions of the top 10 mod-

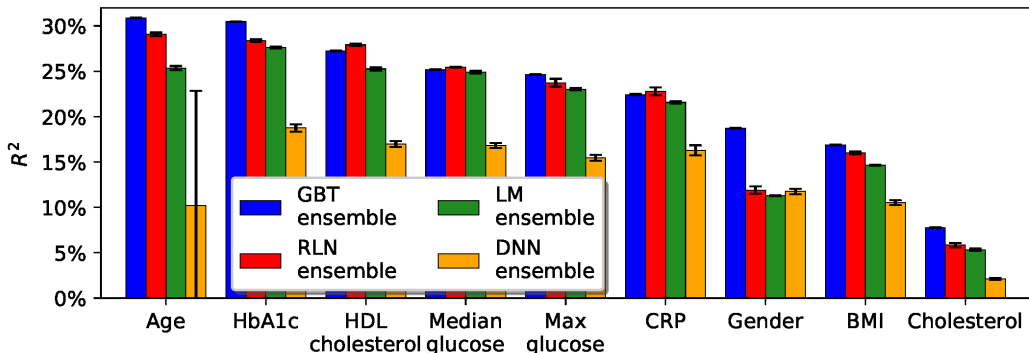

Figure 4: Ensembles of different predictors.

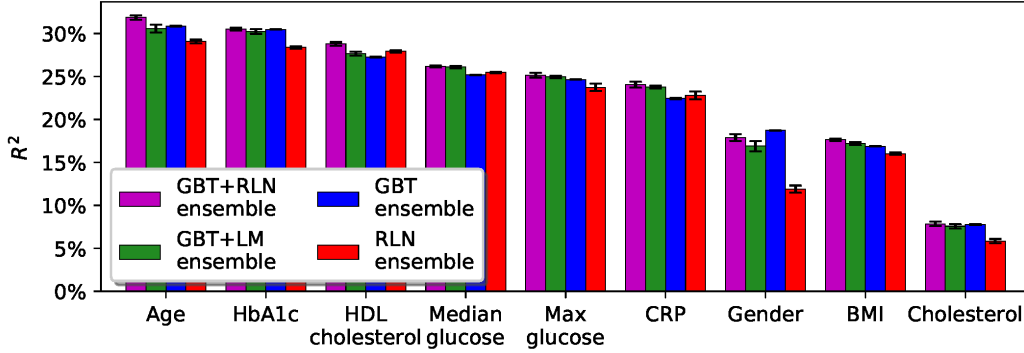

Figure 5: Results of various ensembles that are each composed of different types of models.

| Trait | RLN + GBT | LM + GBT | GBT | RLN | Max |
|---|---|---|---|---|---|
| Age | **31.9% ± 0.2%** | 30.5% ± 0.5% | 30.9% ± 0.1% | 29.1%±0.2% | 31.9% |
| HbA1c | **30.5% ± 0.2%** | 30.2% ± 0.3% | 30.5% ± 0.04% | 28.4%±0.1% | 30.5% |
| HDL cholesterol | **28.8% ± 0.2%** | 27.7% ± 0.2% | 27.2% ± 0.04% | 27.9%±0.1% | 28.8% |
| Median glucose | **26.2% ± 0.1%** | 26.1% ± 0.1% | 25.2% ± 0.04% | 25.5%±0.1% | 26.2% |
| Max glucose | **25.2% ± 0.3%** | 25.0% ± 0.1% | 24.6% ± 0.03% | 23.7%±0.4% | 25.2% |
| CRP | **24.0% ± 0.3%** | 23.7% ± 0.2% | 22.4% ± 0.1% | 22.8%±0.4% | 24.0% |
| Gender | 17.9% ± 0.4% | 16.9% ± 0.6% | **18.7% ± 0.03%** | 11.9%±0.4% | 18.7% |
| BMI | **17.6% ± 0.1%** | 17.2% ± 0.2% | 16.9% ± 0.04% | 16.0%±0.1% | 17.6% |
| Cholesterol | **7.8% ± 0.3%** | 7.6% ± 0.3% | 7.8% ± 0.1% | 5.8% ± 0.2% | 7.8% |

Table 1: Explained variance ($R^2$) of various ensembles with different types of models. Only the 4 ensembles that achieved the best results are shown. The best result for each trait is highlighted, and underlined if it outperforms significantly all other ensembles.

els of LM and GBT is only $0.13\% \pm 0.05\%$ and $0.26\% \pm 0.02\%$, respectively. As expected, the high variance of RLN and DNN models allows ensembles of these models to improve the performance over a single model by $1.5\% \pm 0.7\%$ and $4\% \pm 1\%$ respectively, while LM and GBT only improve by $0.2\% \pm 0.3\%$ and $0.3\% \pm 0.4\%$, respectively. Despite the improvement, DNN ensembles still achieve the worst results on all of the traits except for *Gender* and achieve results $9\% \pm 1\%$ lower than GBT ensembles (Figure 4). In comparison, this improvement allows RLN ensembles to outperform GBT ensembles on *HDL cholesterol, Median glucose,* and *CRP*, and to obtain results $8\%\pm1\%$ higher than DNN ensembles and only $1.4\% \pm 0.1\%$ lower than GBT ensembles.

Using ensemble of different types of models could be even more effective because their errors are likely to be even more uncorrelated than ensembles from one type of model. Indeed, as shown in Figure 5, the best performance is obtained with an ensemble of RLN and GBT, which achieves the best results on all traits except *Gender*, and outperforms all other ensembles significantly on *Age*, *BMI*, and *HDL cholesterol* (Table 1)

## 6    Analysis

We next sought to examine the effect that our new type of regularization has on the learned networks. Strikingly, we found that RLNs are extremely sparse, even compared to $L_1$ regulated networks. To demonstrate this, we took the hyperparameter setting that achieved the best results on the *HbA1c* task for the DNN and RLN models and trained a single network on the entire dataset. Both models achieved their best hyperparameter setting when using $L_1$ regularization. Remarkably, $82\%$ of the

input features in the RLN do not have any non-zero outgoing edges, while all of the input features have at least one non-zero outgoing edge in the DNN (Figure 6a). A possible explanation could be that the RLN was simply trained using a stronger regularization coefficients, and increasing the value of $\lambda$ for the DNN model would result in a similar behavior for the DNN, but in fact the RLN was obtained with an average regularization coefficient of $\theta = -6.6$ while the DNN model was trained using a regularization coefficient of $\lambda = -4.4$. Despite this extreme sparsity, the non zero weights are not particularly small and have a similar distribution as the weights of the DNN (Figure 6b).

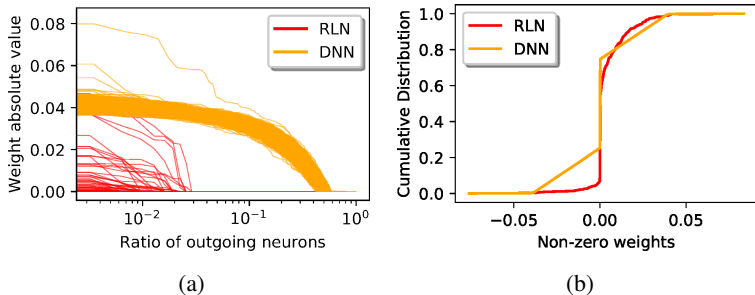

(a)            (b)

We suspect that the combination of a sparse network with large weights allows RLNs to achieve their improved performance, as our dataset includes features with varying relative importance. To show this, we re-optimized the hyperparameters of the DNN and RLN models

Figure 6: a) Each line represents an input feature in a model. The values of each line are the absolute values of its outgoing weights, sorted from greatest to smallest. Noticeably, only $12\%$ of the input features have any non-zero outgoing edge in the RLN model. b) The cumulative distribution of non-zero outgoing weights for the input features for different models. Remarkably, the distribution of non-zero weights is quite similar for the two models.

after removing the covariates from the datasets. The covariates are very important features (Figure 1), and removing them would reduce the variability in relative importance. As can be seen in Figure 7a, even without the covariates, the RLN and GBT ensembles still achieve the best results on 5 out of the 9 traits. However, this improvement is less significant than when adding the covariates, where RLN and GBT ensembles achieve the best results on 8 out of the 9 traits. RLNs still significantly outperform DNNs, achieving explained variance higher by $2\% \pm 1\%$, but this is significantly smaller than the $9\% \pm 2\%$ improvement obtained when adding the covariates (Figure 7b). We speculate that this is because RLNs particularly shine when features have very different relative importances.

To understand what causes this interesting structure, we next explored how the weights in RLNs change during training. During training, each edge performs a traversal in the $w, \lambda$ space. We expect that when $\lambda$ decreases and the regularization is relaxed, the absolute value of $w$ should increase, and vice versa. In Figure 8, we can see that **99.9%** of the edges of the first layer finish the training with a zero value. There are still $434$ non-zero edges in the first layer due to the large size of the network. This is not unique to the first layer, and in fact, **99.8%** of the weights of the entire network have a zero value by the end of the training. The edges of the first layer that end up with a non-zero weight are decreasing rapidly at the beginning of the training because of the regularization, but during the first 10-20 epochs, the network quickly learns better regularization coefficients for its edges. The regularization coefficients are normalized after every update, hence by applying stronger regularization on some edges, the network is allowed to have a more relaxed regularization on other edges and consequently a larger weight. By epoch 20, the edges of the first layer that end up with a non-zero weight have an average regularization coefficient of $-9.4$, which is significantly smaller than their initial value $\theta = -6.6$. These low values pose effectively no regularization, and their weights are updated primarily to minimize the empirical loss component of the loss function, $\mathcal{L}$.

Finally, we reasoned that since RLNs assign non-zero weights to a relatively small number of inputs, they may be used to provide insights into the inputs that the model found to be more important for generating its predictions using Garson's algorithm [10]. There has been important progress in recent years in sample-aware model interpretability techniques in DNNs [28, 31], but tools to produce sample-agnostic model interpretations are lacking [15].[4] Model interpretability is particularly important in our problem for obtaining insights into which bacterial species contribute to predicting each trait.

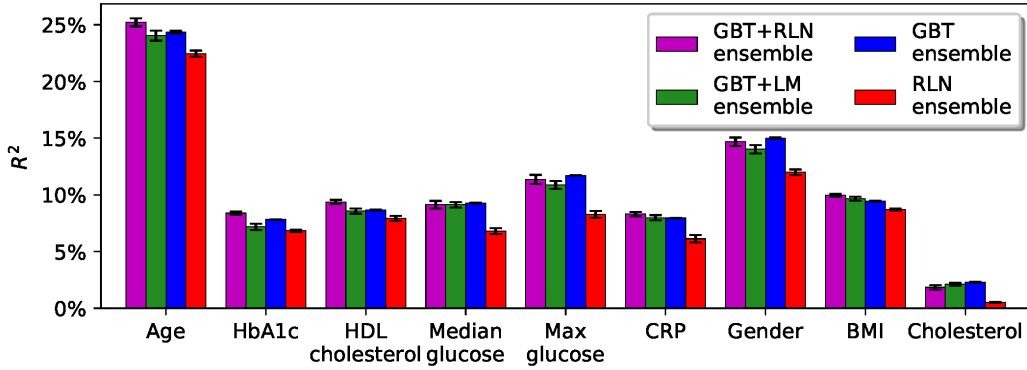

(a)

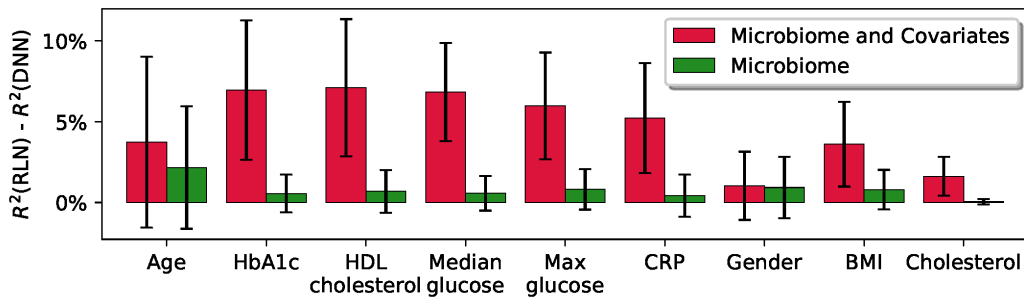

(b)

Figure 7: a) Training our models without adding the covariates. b) The relative improvement RLN achieves compared to DNN for different input features.

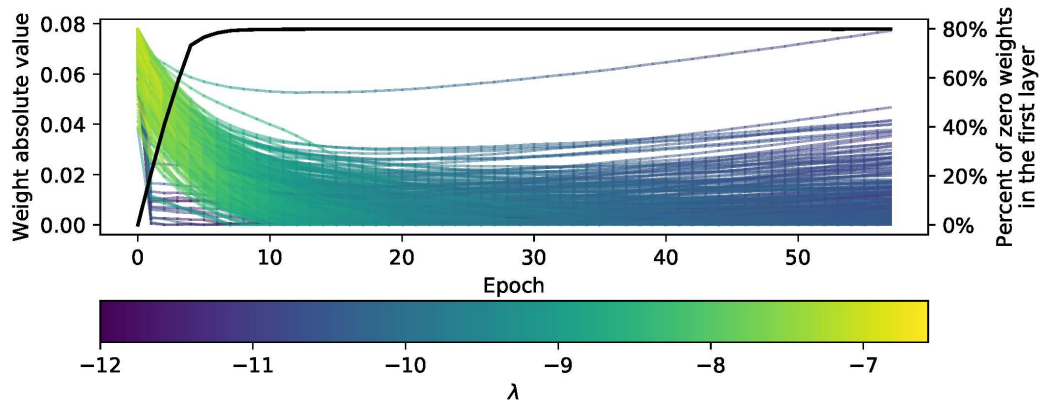

Figure 8: On the left axis, shown is the traversal of edges of the first layer that finished the training with a non-zero weight in the $w$, $\lambda$ space. Each colored line represents an edge, its color represents its regularization, with yellow lines having strong regularization. On the right axis, the black line plots the percent of zero weight edges in the first layer during training.

Evaluating feature importance is difficult, especially in domains in which little is known such as the gut microbiome. One possibility is to examine the information it supplies. In Figure 9a we show the feature importance achieved through this technique using RLNs and DNNs. While the importance in DNNs is almost constant and does not give any meaningful information about the specific importance of the features, the importance in RLNs is much more meaningful, with entropy of the 4.6 bits for the RLN importance, compared to more than twice for the DNN importance, 9.5 bits.

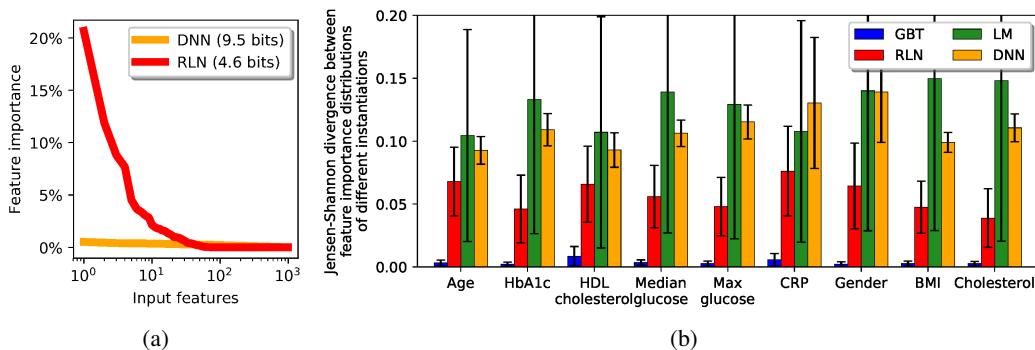

<table>
<tr><td>(a)</td><td>(b)</td></tr>
</table>

Figure 9: a) The input features, sorted by their importance, in a DNN and RLN models. b) The Jensen-Shannon divergence between the feature importance of different instantiations of a model.

Another possibility is to evaluate its consistency across different instantiations of the model. We expect that a good feature importance technique will give similar importance distributions regardless of instantiation. We trained 10 instantiations for each model and phenotype and evaluated their feature importance distributions, for which we calculated the Jensen-Shannon divergence. In Figure 9b we see that RLNs have divergence values $48\% \pm 1\%$ and $54\% \pm 2\%$ lower than DNNs and LMs respectively. This is an indication that Garson's algorithm results in meaningful feature importances in RLNs. We list of the 5 most important bacterial species for different traits in the supplementary material.

## 7 Conclusion

In this paper, we explore the learning of datasets with non-distributed representation, such as tabular datasets. We hypothesize that modular regularization could boost the performance of DNNs on such tabular datasets. We introduce the *Counterfactual Loss*, $\mathcal{L}_{CF}$, and *Regularization Learning Networks* (RLNs) which use the Counterfactual Loss to tune its regularization hyperparameters efficiently during learning together with the learning of the weights of the network.

We test our method on the task of predicting human traits from covariates and microbiome data and show that RLNs significantly and substantially improve the performance over classical DNNs, achieving an increased explained variance by a factor of $2.75 \pm 0.05$ and comparable results with GBTs. The use of ensembles further improves the performance of RLNs, and ensembles of RLN and GBT achieve the best results on all but one of the traits, and outperform significantly any other ensemble not incorporating RLNs on 3 of the traits.

We further explore RLN structure and dynamics and show that RLNs learn extremely sparse networks, eliminating $99.8\%$ of the network edges and $82\%$ of the input features. In our setting, this was achieved in the first 10-20 epochs of training, in which the network learns its regularization. Because of the modularity of the regularization, the remaining edges are virtually not regulated at all, achieving a similar distribution to a DNN. The modular structure of the network is especially beneficial for datasets with high variability in the relative importance of the input features, where RLNs particularly shine compared to DNNs. The sparse structure of RLNs lends itself naturally to model interpretability, which gives meaningful insights into the relation between features and the labels, and may itself serve as a feature selection technique that can have many uses on its own [13].

Besides improving performance on tabular datasets, another important application of RLNs could be learning tasks where there are multiple data sources, one that includes features with high variability in the relative importance, and one which does not. To illustrate this point, consider the problem of detecting pathologies from medical imaging. DNNs achieve impressive results on this task [32], but in real life, the imaging is usually accompanied by a great deal of tabular metadata in the form of the electronic health records of the patient. We would like to use both datasets for prediction, but different models achieve the best results on each part of the data. Currently, there is no simple way to jointly train and combine the models. Having a DNN architecture such as RLN that performs well on tabular data will thus allow us to jointly train a network on both of the datasets natively, and may improve the overall performance.

**Acknowledgments**

We would like to thank Ron Sender, Eran Kotler, Smadar Shilo, Nitzan Artzi, Daniel Greenfeld, Gal Yona, Tomer Levy, Dror Kaufmann, Aviv Netanyahu, Hagai Rossman, Yochai Edlitz, Amir Globerson and Uri Shalit for useful discussions.

## Footnotes

[1] This is not contradictory to the existence of adversarial examples [12], which are able to fool DNNs by changing a small number of input features, but do not actually depict a different object, and generally are not able to fool humans.

[2]The notation for the regularization term is typically $\lambda \cdot \sum_{i=1}^{n} \|w_i\|$. We use the notation $\exp(\lambda) \cdot \sum_{i=1}^{n} \|w_i\|$ to force the coefficients to be positive, to accelerate their optimization and to simplify the calculations shown.

[3]We assume vanilla SGD is used in this analysis for brevity, but the analysis holds for any derivative-based optimization method.

[4]The sparsity of RLNs could be beneficial for sample-aware model interpretability techniques such as [28, 31]. This was not examined in this paper.

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
