[Supplementary Material]

# Regularization Learning Networks: Deep Learning for Tabular Datasets - Supplementary Material

**Ira Shavitt**
Weizmann Institute of Science
irashavitt@gmail.com

**Eran Segal**
Weizmann Institute of Science
eran.segal@weizmann.ac.il

## A  Dynamics of the regularization coefficients at the limit of many training iterations

We try to give some indications as to why the regularization coefficients $\lambda_{t,i}$ tend to decrease at the limit of many training iterations. In Theorem 1, we show that for every weight, there is a region of values for which the regularization coefficients are continuously decreasing when training RLNs.

**Theorem 1.** *Let $W^* = \arg\min \mathcal{L}(Z, W)$ be the optimal weights with respect to the empirical loss. Let $w_i^* \in W^*$ be one of the weights such that $w_i^* \neq 0$. Assume that the loss function $\mathcal{L}$ is continuous. Assume, without loss of generality, that $w_i^* > 0$. Then there exist $b_i \in (0, w_i^*)$ such that if $w_{t,i} \in (b_i, w_i^*)$, $\lambda_{t,i}$ is decreasing, meaning that $\lambda_{t,i} > \lambda_{t+1,i}$.*

*Proof.* For every $w_{t,i} \in (0, w_i^*)$, the gradient of the regularization term is positive, $r_{t,i} = \exp(\lambda_{t,i}) \cdot \frac{\partial \|w_{t,i}\|}{\partial w_{t,i}} > 0$. $\mathcal{L}$ is continuous, so there exist $a_i < w_i^*$ such that for sufficiently small learning rate $\eta$, for every $w_{t,i} \in (a_i, w_i^*)$, $g_{t+1,i} = \frac{\partial \mathcal{L}(Z_{t+1}, W_{t+1})}{\partial w_{t+1,i}} < 0$. Define $b_i = \max\{0, a_i\}$. For every $w_{t,i} \in (b_i, w_i^*)$ we have that $w_{t,i} > 0$ therefore $r_{t,i} > 0$ and $\frac{\partial \mathcal{L}_{CE}}{\partial \lambda_{t,i}} = -\eta \cdot g_{t+1,i} \cdot r_{t,i} > 0$, which will decrease the regularization coefficient

$$\lambda_{t+1,i} = \lambda_{t,i} + \nu \cdot \eta \cdot g_{t+1,i} \cdot r_{t,i} < \lambda_{t,i}$$

$\square$

At the limit of many training iterations, $w_{t,i}$ tends to reach the regions in which $\lambda_{t,i}$ is continuously decreasing. Denote $W_t^\dagger = \left\{ w_{t,i}^\dagger \right\}_{i=1}^n = \arg\min_{W \in \mathcal{W}} \mathcal{L}^\dagger(Z, W, \Lambda_t)$, the optimal weights of the regularized loss, which change over time due to the changes in $\Lambda_t$. $w_{t,i}$ is updated to optimize $\mathcal{L}^\dagger$, therefore $w_{t,i}$ tends to be close to $w_{t,i}^\dagger$ at the limit of many training iterations for sufficiently small learning rate $\eta$.

Denote $\Lambda_{-\infty} = \{-\infty\}_{i=1}^n$, we have that $\arg\min \mathcal{L}^\dagger(Z, W, \Lambda_{-\infty}) = \arg\min \mathcal{L}(Z, W) = W^*$. From [1] we know that $w_{t,i}^\dagger$ is upper hemicontinuous in $\lambda_{t,i}$, so for sufficiently small regularization coefficients, $w_{t,i}^\dagger$ is close to $w_i^*$. We notice that $w_{t,i}^\dagger$ tends to not only be close to $w_i^*$, but also smaller $w_{t,i}^\dagger \leq w_i^*$. In Theorem 2 we give a bound on the norm of $W_t^\dagger$ and in Theorem 3 we show that $w_{t,i}^\dagger$ is strictly decreasing in $\lambda_{t,i}$.

**Theorem 2.** *For every $\Lambda_t$, $\sum_{i=1}^n \exp(\lambda_{t,i}) \cdot \left\| w_{t,i}^\dagger \right\| \leq \sum_{i=1}^n \exp(\lambda_{t,i}) \cdot \|w_i^*\|$.*

*Proof.* Assume to a contradiction that $\sum_{i=1}^{n} \exp\left(\lambda_{t,i}\right) \cdot \left\|w_{t,i}^{\dagger}\right\| > \sum_{i=1}^{n} \exp\left(\lambda_{t,i}\right) \cdot \|w_i^*\|$, then

$$\mathcal{L}^{\dagger}\left(Z, W^*, \Lambda_t\right) = \mathcal{L}\left(Z, W^*\right) + \sum_{i=1}^{n} \exp\left(\lambda_{t,i}\right) \cdot \|w_i^*\| \leq$$

$$\leq \mathcal{L}\left(Z, W_t^{\dagger}\right) + \sum_{i=1}^{n} \exp\left(\lambda_{t,i}\right) \cdot \left\|w_{t,i}^{\dagger}\right\| = \mathcal{L}^{\dagger}\left(Z, W_t^{\dagger}, \Lambda_t\right)$$

Contradicting the definition of $W_t^{\dagger}$. $\qquad\square$

**Theorem 3.** *Let $i$ be some edge, and let $\Lambda_1 = \{\lambda_{1,j}\}_{j=1}^{n}$, and $\Lambda_2 = \{\lambda_{2,j}\}_{j=1}^{n}$, such that*
$\begin{cases} \lambda_{1,j} = \lambda_{2,j} & j \neq i \\ \lambda_{1,i} < \lambda_{2,i} & j = i \end{cases}$. *Let $W_1^{\dagger} = \arg\min_W \mathcal{L}^{\dagger}\left(Z, W, \Lambda_1\right)$ and $W_2^{\dagger} = \arg\min_W \mathcal{L}^{\dagger}\left(Z, W, \Lambda_2\right)$*
*be the optimal weights when regulating with $\Lambda_1$ and $\Lambda_2$, respectively. Then $\left\|w_{2,i}^{\dagger}\right\| < \left\|w_{1,i}^{\dagger}\right\|$.*

*Proof.* Assume to a contradiction that $\left\|w_{2,i}^{\dagger}\right\| \geq \left\|w_{1,i}^{\dagger}\right\|$, then

$$\mathcal{L}^{\dagger}\left(Z, W_1^{\dagger}, \Lambda_2\right) = \mathcal{L}\left(Z, W_1^{\dagger}\right) + \sum_{i=1}^{n} \exp\left(\lambda_{2,i}\right) \cdot \left\|w_{1,i}^{\dagger}\right\| <$$

$$< \mathcal{L}\left(Z, W_1^{\dagger}\right) + \sum_{i=1}^{n} \exp\left(\lambda_{1,i}\right) \cdot \left\|w_{1,i}^{\dagger}\right\| =$$

$$= \mathcal{L}^{\dagger}\left(Z, W_1^{\dagger}, \Lambda_1\right) \leq \mathcal{L}^{\dagger}\left(Z, W_2^{\dagger}, \Lambda_1\right) =$$

$$= \mathcal{L}\left(Z, W_2^{\dagger}\right) + \sum_{i=1}^{n} \exp\left(\lambda_{1,i}\right) \cdot \left\|w_{2,i}^{\dagger}\right\| <$$

$$< \mathcal{L}\left(Z, W_2^{\dagger}\right) + \sum_{i=1}^{n} \exp\left(\lambda_{2,i}\right) \cdot \left\|w_{2,i}^{\dagger}\right\| =$$

$$\mathcal{L}^{\dagger}\left(Z, W_2^{\dagger}, \Lambda_2\right)$$

Contradicting the definition of $W_2^{\dagger}$. $\qquad\square$

The increase in the other regularization coefficients $\lambda_{t,j}$ for $j \neq i$ could increase $w_{t,i}^{\dagger}$. Theorem 4 shows that even if $w_{t,i}^{\dagger} > w_i^*$, the Regularization Learning will tend to decrease $w_{t,i}^{\dagger}$ back.

**Theorem 4.** *If $\mathcal{L}$ is continuous, for sufficiently small learning rate $\eta$ and regularization coefficients $\Lambda_t$ and large enough $t$, if $w_{t,i}^{\dagger} > w_i^*$, then $w_{t,i}^{\dagger}$ is decreasing in $t$, meaning that $w_{t,i}^{\dagger} > w_{t+1,i}^{\dagger}$.*

*Proof.* $\mathcal{L}$ is continuous, then there's a neighborhood $(w_i^*, c_i)$ such that for sufficiently small learning rate $\eta$, for every $t$ such that $w_{t,i} \in (w_i^*, c_i)$, $g_{t+1,i} = \frac{\partial \mathcal{L}(Z_{t+1}, W_{t+1})}{\partial w_{t+1,i}} > 0$. For sufficiently small regularization coefficients, $w_{t,i}^{\dagger}$ is not too far $w_i^*$, and for large enough $t$ and small enough learning rate $\eta$, $w_{t,i}$ is not too far from $w_{t,i}^{\dagger}$, such that $w_{t,i} \in (w_i^*, c_i)$. $r_{t,i} > 0$ since $w_{t,i} > 0$, giving $\frac{\partial \mathcal{L}_{CE}}{\partial \lambda_{t,i}} = -\eta \cdot g_{t+1,i} \cdot r_{t,i} < 0$, which will increase the regularization coefficient $\lambda_{t,i} < \lambda_{t+1,i}$. $w_{t,i}^{\dagger}$ strictly decreases with $\lambda_{t,i}$, giving $w_{t,i}^{\dagger} > w_{t+1,i}^{\dagger}$. $\qquad\square$

Theorem 4 show that at the limit of many training iterations, if $w_{t,i}^{\dagger} > w_i^*$ then $w_{t,i}^{\dagger}$ will tend to decrease in training, and Theorem 1 shows that there is a region $w_{t,i}^{\dagger} \in (b_i, w_i^*)$ in which $\lambda_{t,i}$ is continuously decreasing, which might give some indication as to why $\lambda_{t,i}$ tends to decrease at the limit of many training iterations.

## B  Regularization Learning algorithm

**Require:** $Z = \{(x,y)_m\}_{m=1}^M$, $W = \{w_i\}_{i=1}^n$, $\Lambda = \{\lambda_i\}_{i=1}^n$, $\nu, \eta, \theta$
1: **for** $i = 1$ to $n$ **do**
2:     $r_{prev,i} \leftarrow null$
3: **end for**
4: **for** $t = 1$ to $T$ **do**
5:     **for** $i = 1$ to $n$ **do**
6:         $g_i \leftarrow \frac{\partial \mathcal{L}(Z,W)}{\partial w_i}$
7:         **if** $r_{prev,i} \neq null$ **then**
8:             $\lambda_i \leftarrow \lambda_i + \nu \cdot \eta \cdot g_i \cdot r_{prev,i}$
9:         **end if**
10:    **end for**
11:    **for** $i = 1$ to $n$ **do**
12:        $\lambda_i \leftarrow \lambda_i + \left(\theta - \frac{\sum_{i=1}^n \lambda_i}{n}\right)$
13:        $r_i \leftarrow \exp(\lambda_i) \cdot \frac{\partial \|w_i\|}{\partial w_i}$
14:        $w_i \leftarrow w_i - \eta \cdot (g_i + r_i)$
15:        $r_{prev,i} \leftarrow r_i$
16:    **end for**
17: **end for**
18: **return** $W$

**Algorithm 1:** Regularization Learning

## C  Microbiome input features and predicted traits

Table 1: Predicted traits

| Trait | Description |
|---|---|
| Age | |
| HbA1c | Glycated hemoglobin, a marker for cardiovascular disease, used for diabetes diagnosis |
| HDL Cholesterol | High-density lipoprotein cholesterol, related to cardiovascular health |
| Median Glucose | The median blood glucose level of the patient measured during a week |
| Max glucose | The maximal blood glucose level of the patient measured during a week |
| CRP | C-reactive protein, a marker of inflammation |
| Gender | |
| BMI | Body-Mass Index, used to estimate adiposity, a risk factor for numerous disease |
| Cholesterol | Risk factor for cardiovascular disease |

The full list of the input features are as follows:

1. Covariates: age, gender, and BMI of the person.

2. Microbiome data:

   (a) The log relative abundance of all bacterial species

   (b) The 100 first components of the Principal Component Analysis (PCA) of the matrix of log relative abundance of species

   (c) The 100 first components of the Principal Component Analysis (PCA) of the matrix of log relative abundance of genes

   (d) Microbiome metadata:

      i. The fraction of reads that were mapped to bacterial species, bacterial genes, and human genes, out of all the reads

      ii. The kit that was used to collect the sample

    iii. The entropy of relative abundances of the 10, 20, 50 and 100 most prevalent species

    iv. The number of different species with abundance greater than $10^{-3}$, $10^{-4}$, $10^{-5}$ (the alpha diversity)

For each of the traits we took the $5-95$ percentiles of its values and trimmed the values of each person to be within 3 standard deviations (STDs) of the mean of these $5-95$ percentile values. Before training the models, the features and the labels were standardized.

The basic covariates are also some of the traits used as the prediction labels. When training a model to predict one of the covariates, the respective covariate was not used as an input features.

## D   Hyperparameters and training settings

The hyperparameters of the different models are as follows:

- DNN and RLN models have the following hyperparameters:
  - Number of iterations
  - Learning rate
  - Activation function (ReLU, SoftPlus or tanh)
  - Batch size
  - The number of layers in the network
  - The width of the last layer before the output layer in the network
  - RLN models have the following additional hyperparameters:
    * Regularization norm: ($L_1$ or $L_2$)
    * The average regularization coefficient ($\theta$ in the paper)
    * The learning rate for the regularization coefficients ($\nu$ in the paper)
  - DNN models have the following additional hyperparameters:
    * Type of regularization (dropout, $L_1$, and $L_2$) and its term
- GBT models have the following hyperparameters:
  - Number of trees
  - Learning rate
  - Maximal depth of trees
  - Minimum loss reduction required to make a further partition on a leaf node of the tree ($\gamma$ in *XGBoost*)
  - Subsample ratio of columns when constructing each tree
  - The minimum sum of instance weight (hessian) needed in a child
  - Maximum delta step for the weight estimation of each tree
  - Subsample ratio of the training instance when constructing each tree
  - $L_1$ regularization term
  - $L_2$ regularization term
- LM models have the following hyperparameters:
  - $L_1$ regularization term
  - $L_2$ regularization term
  - Maximal number of iterations
  - Optimization tolerance

For all instantiations of DNN and RLN models, the Glorot [3] normalized initializer and the rmsprop [4] optimizer were used. The layers of the DNNs and RLNs are all fully connected. The width of the last layer before the output layer in the network and the depth of the networks are hyperparameters. The widths of the rest of the layers were calculated to form a geometric series, given that we know the input dimension, the depth, and the width of the last layer before the output layer.

When training RLNs, we initialized the regularization coefficients to be equal, and have the same value as the average regularization coefficient, i.e., $\Lambda_0 = \{\theta\}_{i=1}^n$.

The loss function was the $L_2$ loss, $\mathcal{L}(x,y) = (f(x) - y)^2$.

For some participants, the microbiome of was sampled several times, in which case we down-weighted each of their samples such that the total weight of all samples of each participant was equal to 1.

The hyperparameters for each instantiation were sampled from a hand-tuned distribution over the values of the hyperparameters. From the optimized distributions over the hyperparameters, 50 samples were obtained for each model. We ran our experiments using 10-fold train-test splits of our dataset, with $20\%$ of the training samples being held out as a validation set. The variance of the scores of the different hyperparameters samples are shown in the error bars.

When training ensembles, 30 model instantiations were randomly chosen based and their predictions were averaged. When computing ensembles of several models, 30 random model instantiations for all the types of models in the ensemble were averaged. The variance of the scores of the different instantiations samples are shown in the error bars. The performance of the ensembles plateaued after 30 instantiations, therefore we only present results for ensembles with 30 instantiations per model for consistency.

## E   RLN feature importance for the microbiome dataset

The 5 most important bacterial species for the predicted traits, based on the "feature importance" proposed in [2], are shown in Table 2.

Table 2: The 5 most important microbiome features for different traits

|  | Klebsiella pneumoniae | Streptococcus vestibularis | Bacteroide pectinophilus | Dialister succinatiphilus | Megamonas rupellensis |
|---|---|---|---|---|---|
| Age | 0.85% | 0.77% | 0.66% | 0.43% | 0.55% |
| HbA1c | 1.04% | 0.04% | 0.37% | 0.45% | 0.35% |
| HDL cholesterol | 0.56% | 0.61% | 0.90% | 0.05% | 1.18% |
| Median glucose | 0.31% | 0.53% | 0.87% | 0.03% | 0.47% |
| Max glucose | 0.34% | 0.37% | 0.07% | 0.55% | 0.56% |
| CRP | 0.20% | 0.84% | 0.16% | 1.02% | 0.30% |
| Gender | 0.71% | 0.30% | 1.07% | 0.28% | 0.12% |
| BMI | 0.95% | 0.86% | 0.26% | 1.18% | 0.87% |
| Cholesterol | 0.54% | 0.78% | 0.55% | 0.53% | 0.00% |