[Reviews · NeurIPS 2018]

Reviewer 1



The manuscript “Regularization Learning Networks” introduces a new type of neural network, trained using a new loss function, called Counterfactual Loss, together with stochastic gradient descent. The main hypothesis of the manuscript is that more flexibility in regularization should enable a better treatment of features that have a varying degree of importance. With this the authors wanted to enable DNNs to perform better in scenarios where they currently loose against other learning methods like gradient boosting trees. The authors motivate in their conclusion why this might make sense, but in the application they show it is not clear, why one has to use a regularization learning network (RLN) instead of a GBT model. Quality The overall quality of the paper is good. The authors really tried to show the effects of the new regularization scheme, which seems to produce reasonable results. Clarity The presentation of the method is very clear. Sometimes, it would have been nice to include more details in the manuscript and not in the supplement (e.g., training setup). Significance The regularization scheme seems to have certain benefits over standard DNNs. The authors could not show on the data set they chose that this leads to a model with better performance than the best other method, but at least compared to standard DNNs the performance increase was significant. Therefore, especially for settings where joint models on several types of data are supposed to be learnt, this new approach might be a good way forward. Since the authors only gave this application as an outlook and also only mention that the initial results are positive (in the author response), it remains to be seen how well the performance will be for different prediction tasks of this type.

Reviewer 2



The Authors describe a way to incorporate unequal regularization into training methods of deep learning training in a way that does now spawn a nan value parameters to optimize . I think this paper is of fundamental importance not just from the methodological point of view, presenting a truly new method, but from the fact that it addresses tabular data. Something scarcely done, such it deserves a strong accept The main point is that by coupling a minimization scheme for a “sudo” loss one can contain the increasing number of parameters. I would be very interested in how does decay of the counterfactual loss over time scale with the number of parameters, is there some of scaling function ? will it be a power law ? In general I am wondering if the authors performed some sort of dynamical analysis to the counterfactual loss, my intuition tells me that the way the problem is stated there is a shadow lemma and the counterfactual loss mimics some sort of Nekrasov bounded system and thus most of the parameters don’t really come into play. For full disclosure out of curiosity I have implemented and recreated the experiments in pytorch/mxnet/tensorflow, keras and I did not get the exact results but slight deviations from the results the authors report (which are still very very good). Can authors give some more details into the experiments ? In general results varied between only slightly between different platforms and I am keen to know how exactly did the authors achieve their results, will there be a code publically available somewhere ?

Reviewer 3



Summary This work develops Regularization Learning Networks (RLN), an approach to learn varying levels of regularization for different network weights. Regularization levels for (D)NN are usually a global hyper-parameter, optimized via CV for the whole network. The author point out that figuring different levels or regularization for different features is appealing especially for data where features are of different “type” with some possibly highly important while others not and the transition between those is more striking than common in say structured images. Good examples are smaller tabular data of expression, EHR where DL commonly underperforms. The CV approach without a direct target function to optimize does not scale for optimizing each weight separately. Instead, the authors develop an elegant solution in the paper, the counterfactual loss (L_cf). The SGD now has two contributions: the gradient for the loss and the gradient for the regularization, as a result of evaluating the delta from moving the weight i to a new point W_{t+1,i} on a new batch Z_{t+1} as a derivative of the hyperparameter for penalty \Lamba_i. The authors evaluate the performance of their method on a dataset of microbiome from patients with several different target values to predict (Age, BMI, etc.). Their RLN greatly improves compared to DNN with standard “global” regularization (optimized by CV), but GBT is still better. They also try ensembles of learners. As expected these give improved results, with the best predictor composed of a combination of GBT and RLN. Finally, they evaluate the level or regularization on the features during learning (Fig7) and the features importance compared to a DNN (Fig8), showing learned models are extremely sparse. Pros: The paper addresses an important and challenging problem. The solution seems novel, simple/elegant, yet efficient. Paper is generally clearly written. The solution appears to significantly improve performance compared to standard DNN regularization. It was nice to see the authors also make an effort to understand “what is going on” in terms of regularization (Fig7) and interpretation applying matrix product (p. 8 top paragraph, Fig 8). Cons: Overall, we did not find any major flaws though several improvements can definitely be made. First, we note that at the end of the day on all tasks tested GBT was still the better model. Only an ensemble of GBT+ RLN gave a very slight improvement. Thus, on all tasks tested, there is no real benefit in the proposed method. The above is related to the fact that while the authors rightfully motivate their work with a general hypothesis about (un)structured/tabular data etc. they only use a single dataset with no synthetic testing to explore behavior. Why? We would like to see the final including a more elaborate evaluation of different datasets which the authors are already aware of (e.g. line 21). This could include different synthetic settings. For example: What happens if the features are informative but (highly) correlated? How does the amount of data affect the new framework? Even on the dataset used the effect of the RLN specific hyperparameters are not explored (they are listed in the supp). Similarly, the analysis of feature importance and how to decide on those important ones was lacking. Why are 5 features selected? What is the criteria for significance? Do they make sense? There are many other ways to extract meaning from DNN (e.g. IG) which are at least worth discussing. It is also important that the authors make code/analysis available - it could be highly useful *if* indeed it proves to give a boost (pun intended) in performance in other domains. Other comments: In the introduction and related work the authors keep interchanging the (undefined) terms (un)structured data, (non) distributed representation, and tabular data. Though the general intention is clear this adds unnecessary confusion. The authors can do a better job introducing, deriving the main idea (p.3) and giving the intuition behind it. As is, it took a few back and forth to understand things. Many of the figures are informative yet not so user-friendly and their caption limited (Fig1, 5, 7, 8). Consider plotting differently or adding info.